# The Role of Extracellular Vesicles in Allergic Sensitization: A Systematic Review

**DOI:** 10.3390/ijms25084492

**Published:** 2024-04-19

**Authors:** Davis Tucis, Georgina Hopkins, William Browne, Victoria James, David Onion, Lucy C. Fairclough

**Affiliations:** 1School of Life Sciences, The University of Nottingham, Nottingham NG7 2UH, UK; davis.tucis@nottingham.ac.uk (D.T.); georgina.hopkins@nottingham.ac.uk (G.H.); msxwb2@nottingham.ac.uk (W.B.); david.onion@nottingham.ac.uk (D.O.); 2School of Veterinary Medicine and Science, The University of Nottingham, Nottingham NG7 2UH, UK; victoria.james@nottingham.ac.uk

**Keywords:** allergy, extracellular vesicles, sensitization, tolerance, Th2

## Abstract

Allergies affect approximately 10–30% of people worldwide, with an increasing number of cases each year; however, the underlying mechanisms are still poorly understood. In recent years, extracellular vesicles (EVs) have been suggested to play a role in allergic sensitization and skew to a T helper type 2 (Th2) response. The aim of this review is to highlight the existing evidence of EV involvement in allergies. A total of 22 studies were reviewed; 12 studies showed EVs can influence a Th2 response, while 10 studies found EVs promoted a Th1 or Treg response. EVs can drive allergic sensitization through up-regulation of pro-Th2 cytokines, such as IL-4 and IL-13. In addition, EVs from MRSA can induce IgE hypersensitivity in mice towards MRSA. On the other hand, EVs can induce tolerance in the immune system; for example, pre-exposing OVA-loaded EVs prevented OVA sensitization in mice. The current literature thus suggests that EVs play an essential role in allergy. Further research utilizing human in vitro models and clinical studies is needed to give a reliable account of the role of EVs in allergy.

## 1. Introduction

Atopic allergies, also known as immediate or type I hypersensitivities, affect between 10% and 30% of the population [1]. Allergies affect people’s quality of life, impacting their work and personal lives; in addition, the treatments for allergies bear a heavy burden on healthcare systems across the globe. It has been reported that it costs the NHS over GBP one billion per annum [2]. However, the underlying mechanism of how an allergy develops is still poorly understood. The most frequent clinical manifestations of allergies are asthma and allergic rhinitis, but other atopic diseases include urticaria and anaphylactic shock.

Several factors can impact whether someone develops atopic allergy, such as environmental exposure, genetic predispositions or lifestyle factors, such as a Westernized diet, low vitamin D intake, antibiotic use, and the route of birth; all these form an intricate balance that determines the development of type 1 hypersensitivity [3].

Allergic sensitization, the first phase of atopic allergy development, is central to the initiation of atopic diseases. Typically, proteins within allergen sources initiate allergic sensitization through recognition by dendritic cells (DCs) and presentation to naïve T cells, which results in a cascade of reactions shifting the response in a Th2-dominant manner. Dendritic cells can be classified into four subsets—human plasmacytoid DC (pDC), conventional DC1 (cDC1), conventional DC2 (cDC2) and monocyte-derived DC (mo-DC) [4]. Subsequently, allergen-specific immunoglobulin E (IgE) is bound on mast cells through the FcεRI receptor. Conversely, naïve T cells can differentiate into Th1 or Treg cells, resulting in immune tolerance, which normally happens in healthy individuals. Recent research has suggested other components can drive allergic sensitization, such as lipids [5,6], carbohydrates [7], and most recently, EVs, which can, in fact, also carry proteins, lipids and carbohydrates [8,9].

EVs are small lipid-bilayer particles that are released naturally from cells within the body and by other sources, such as bacteria. They are key inter-cellular communicators, and their size can range from 20 nm to 10 µm [10]. EVs can transport proteins, nucleic acids (deoxyribonucleic acid (DNA), messenger ribonucleic acid (mRNA), and non-coding RNA), lipids, metabolites and specific cell surface markers depending on the cells from which they originate [11]. They influence recipient cells due to their capacity to transfer this ‘cargo’ [12]. When discriminating between EVs, their origin is considered; if they develop internally and are then released through the plasma membrane, they are called exosomes, where their size is usually smaller than 200 nm. If they bud off from the plasma membrane, they are called ectosomes, where their size is 100 nm to 10 µm. The recommendations for carrying out research into EVs are formulated in Minimal Information For Studies of Extracellular Vesicles (MISEV) 2023 [13].

Importantly, EVs and their contents represent the cell they come from, as well as the current condition of the cell [14]. Due to their ability to reflect the condition of their cellular origin, they are often used as biomarkers for diseases; however, studies have also examined their involvement in the pathogenesis of diseases [14]. Indeed, recent studies have been trying to understand the role of EVs in allergy. For example, in a study done by Fang et al. in 2020, it was found that by utilizing human mesenchymal stem cell (MSC) EVs, it was possible to alleviate inflammation in the lungs by restricting the infiltration of inflammatory cells and epithelial goblet cells by changing recruitment and polarization of the alveolar macrophages [15]. In asthma, EVs can be used as a diagnostic tool between healthy and asthmatic patients, and EVs can promote inflammation and airway remodeling, suggesting EVs also play a role in the pathogenesis of asthma, which has been systematically reviewed in a study done by Sangaphunchai et al., 2020 [16].

In this systematic review, the focus is on the role of EVs in allergic sensitization. The role of EVs generated from the host (termed Host-derived EVs) and from exogenous sources, such as bacteria (termed exogenous EVs), is discussed, and whether these EVs induce sensitization or tolerance.

## 2. Materials and Methods

### 2.1. Search Strategy

This review was conducted with the implementation of the Preferred Reporting Items for Systematic Reviews and Meta-Analyses (PRISMA) guidelines. The search strategy was constructed around two key terms: ‘extracellular vesicles’ and ‘allergy’. In this review, two databases were searched to find relevant papers: PubMed and Embase, from their inception to 31 December 2023, using the following search terms:

PubMed—(exosomes or exosome* or extracellular vesicle* or ectosome or microparticle or microvesicle or shedding vesicle*) AND (allergic sens* or allergic sensitization or type 1 hypersensitivity or allergy)

EMBASE—(exosomes or exosome* or extracellular vesicle* or ectosome or microparticle or microvesicle or shedding vesicle*) AND (allergic sens* or allergic sensitization or type 1 hypersensitivity or allergy).mp. [mp = title, abstract, heading word, drug trade name, original title, device manufacturer, drug manufacturer, device trade name, keyword, floating subheading word, candidate term word].

### 2.2. Selection of Studies

This systematic review aimed to examine the evidence on the role and function of EVs in allergic sensitization, with the hypothesis that EVs from various origins contribute to the development of allergy. For this review, other reviews, books, editorials, and conference proceedings were excluded, language was restricted to English, and there was no restriction on the date of publication. Publications were screened for their relevance against pre-defined inclusion and exclusion criteria (Table 1), and a random subset (10%) of these studies was reviewed independently by a second reviewer.

### 2.3. Study Quality Assessment

To assess the robustness of the studies included in this systematic review and to highlight the current research methodology in this field, a scoring system was developed, with specific criteria set in Table 2. Each study’s score was assessed by the criteria: model, robustness of model, sample size, EV isolation, EV characterization, and sensitization method. Scores from the studies were summed and divided by the highest possible total score to calculate the study bias score. Studies that received lower scores are not less reliable but are missing key methods in researching EVs, such as ensuring pure and intact EV populations through isolation.

## 3. Results

### 3.1. PRISMA Search and Selection

A PRISMA diagram was used to track the search details for this systematic review (Figure 1). The search resulted in 1356 papers: 1044 from PubMed and 312 from EMBASE. After the removal of duplicated publications, 1271 titles and abstracts of papers were examined for relevance. After the title and abstract review, 336 papers were kept as potential interest for the review. After selecting papers using the inclusion and exclusion criteria, 22 studies were included in the systematic review, which were relevant to understanding the role of EVs in allergic sensitization.

The first publication relevant to EVs in allergic sensitization was published in 2007; however, 60% of the papers were published in the last 8 years (Figure 2A).

The results highlight two main types of EVs based on their source of origin: EVs that were produced from the host (host-derived) or from external sources (exogenous), such as bacteria (Figure 2B). Thus, the results will be discussed in terms of host-derived or exogenous EVs and their potential to either promote allergic sensitization or the tolerance of allergens.

### 3.2. Host-Derived Extracellular Vesicles

#### 3.2.1. Promotion of Allergic Sensitization

In the allergic sensitization process, naïve T cells differentiate into Th2 cells, resulting in interleukin 4 (IL-4) production and IgE deposition on mast cells, but what drives this shift to Th2 is poorly understood. Publications suggesting that host-derived EVs can promote Th2 differentiation and, thus, the sensitizations were identified and summarized in Table 3.

Seven studies examined the role of EVs in promoting Th2-like cytokine production and inducing the differentiation to Th2 cells [17,18,19,20,21,22,23]. B-cell-derived EVs had allergen peptide-loaded major histocompatibility complex (MHC) molecules on their surface and were capable of stimulating T cells to proliferate and produce IL-5 and IL-13 [17]. A different study observed that plasma EVs have antigen-presenting capabilities and induced Th2 cell differentiation and increased levels of IL-13 [18]. One in vitro study found that thymic stromal lymphopoietin (TLSP)-activated DC EVs induced a higher number of Th2 cells by elevating IL-4 concentration and decreasing interferon-gamma (IFN-γ) concentration, thus reducing Th1 cells [19]. Investigating the effect of T follicular helper cell (Tfh) EVs found that these EVs could regulate DC maturation through miR-142-5p/CDK5/STAT3 pathway; however, most importantly, Tfh EVs promoted Th2 responses in naïve mice through elevated levels of IL-13 and IgE, while simultaneously decreasing IFN-γ and IL-17 levels [20]. Another study showed that DC-derived EVs could carry Fel d 1 and in co-culture with peripheral blood mononuclear cells (PBMCs) induced higher levels of IL-4 production [21]. By researching lncRNA Nuclear Paraspeckle Assembly Transcript 1 (NEAT1) it was found that it was responsible for the promotion of allergic rhinitis (AR) pathogenesis via EVs, which interacted with human nasal epithelial cells (HNECs) and induced IL-13 production and apoptosis, which can lead to damage to the epithelium and increase the likelihood of allergen exposure and development of allergic sensitization [22]. Epithelium-derived EVs carried long non-coding RNA growth arrest-specific 5 (GAS5), which can suppress Th1 differentiation by downregulating the enhancer of zeste 2 polycomb repressive complex 2 subunit (EZH2) and T-bet, where IFN-γ concentration and Th1 responses were decreased, while also stimulating IL-4 production and Th2 differentiation [23]. 

Of the seven papers describing host-derived EVs driving allergic sensitization, all studies used human samples for the isolation of EVs and applied them in an in vitro model [17,18,19,20,21,22,23], whereas one study also used mice and an in vivo model [20]. Regarding the EVs isolation methodology, six studies used serial ultracentrifugation [17,18,19,20,21,23], while one study used the ExoQuick precipitation kit, which utilized size exclusion chromatography (SEC) [22]. The identification process was different across the studies; however, six studies used TEM to detect the presence of EVs [18,19,20,21,22,23], and four studies used Western blotting to confirm the presence of tetraspanins [18,19,22,23]. Three papers used flow cytometry to detect tetraspanins on the EVs [17,18,21]. Two studies used NTA to measure the size of EVs [18,21]. The papers were scored based on their quality using specific criteria created for this systematic review (Table 2). Overall, the identification of EVs was sufficient in three papers [18,19,21] as two or more methods were used, while in the other papers [17,20,22,23] where only one identification method was used (TEM). Two papers received high scores of 63% [18,21] due to their thorough characterization of EVs, while three studies had the lowest scores of 52% [17,19,23] because of the limited work done in EV isolation and characterization. A full assessment of the papers is presented in Table 4.

Overall, all seven studies show the potential of EVs influencing the immune system to shift towards a Th2 response by increasing the differentiation towards Th2 cells and Th2-like cytokine production, which could potentially reveal how EVs influence allergic sensitization. By understanding allergy development, it may be possible to prevent its occurrence.

#### 3.2.2. Promotion of Immune Tolerance

Naïve CD4 T-cells can differentiate into Th1, Th2 or Treg cells; if it is a Th1- or Treg-dominant response, immune tolerance is promoted. Papers suggesting that EVs can drive immune tolerance are compiled in Table 5.

Nine studies suggested that host-derived EVs were capable of suppressing Th2 responses, resulting in immune tolerance [24,25,26,27,28,29,30,31,32]. Four of these studies researched the influence of EVs on food allergy [24,25,26,27], one of which found intestinal epithelial-cell-derived integrin ɑβ6 and OVA-carrying EVs interacting with T cells could induce T cell differentiation to Treg cells, through the increased production of TGF-β [24]. Furthermore, EVs from IL-2 and OVA-stimulated DCs, with MHC class II molecules, interacting with OVA-specific CD4^+^ T cells induced differentiation into Tregs [25]. Another study relating to food allergy showed that EVs carry the allergen-MHC class 2 complexes and IL-10 and could induce Tr1 differentiation in healthy CD4+ T cells [26]. Another study found that MSC-derived EVs containing OVA pre-exposure in mice resulted in tolerance to OVA, with decreased levels of IgE and IL-4, along with elevated TGF-β levels [27].

Three studies examined the role of EVs in allergic rhinitis [28,29,30]. In these studies, it was found that when using EVs, it was possible to inhibit the Th2 cell responses and induce Treg differentiation by limiting the production of IgE and increased expression of TGF-β and IgG2a [29]. Another study showed that MSC-derived EVs also suppressed the Th2 response by reducing the production of IL-4, IL-9, and IL-13 via IL-10 production, thus increasing the levels of Tregs in the system [28]. The last allergic rhinitis study showed that human mesenchymal stem cells (HMSC)-EVs co-cultured with CD4^+^ T cells inhibited the Th2 differentiation by regulating the miR-146a-5p/SERPINB2 pathway [30]. 

Two studies looked at the role of allergens and EVs in immune tolerance; one of the studies looked at the differences between microvesicles and exosomes and their capacity to influence the immune responses, where it was found that exosomes induced a stronger response, which generated a Th1-like response by inducing the production of allergen-specific IgG abs and expansion of OVA-specific CD8^+^ T cell population [31]. The other study observed the potential of ‘bystander’ tolerance of allergens by EVs, where it was found that EVs inhibited the immune responses to the specific allergen in addition to another unrelated allergen by limiting the production of IL-5 and IL-13 and reducing the inflammatory responses in the lung [32]. 

Of the nine papers describing host-derived EVs in immune tolerance, seven studies used mice as their source of EVs [24,25,26,29,31,32], and two studies used human samples and human cell lines as their EV source [28,30]. EV isolation protocols were similar across the papers, where seven of the studies used some form of serial ultracentrifugation [24,25,26,29,30,31,32], one study used anion exchange chromatography [28], and one study utilized an Exospin isolation kit, which uses SEC [27]. Various methods were used to confirm the presence of isolated EVs; four studies used the Bradford assay [24,25,26,28], five studies used TEM [24,28,29,30,31], and one used SEM [27]. Two studies used Western blotting to detect the presence of tetraspanins [28,30], and one study used immunophenotyping [27]. Overall, the identification and characterization of EVs in these studies were lacking. The highest bias score was 63%, but this study lacked the use of multiple models [28], whilst the lowest score achieved was 44%, resulting from the limited use of models and inferior isolation technique [29]. Six out of nine studies scored over 50% [25,26,28,30,31,32]. The overall scores were compiled into a table below (Table 6).

Overall, these nine studies showed that EVs were capable of suppressing Th2 immune response and skewing towards Th1 responses, as well as the induction of Treg cells, which in turn reduce the inflammatory cytokine and IgE production in food allergy and allergic rhinitis. Thus, these results show the potential of using EVs as a vehicle for alleviating and/or preventing allergies.

### 3.3. Exogenous Extracellular Vesicles

#### 3.3.1. Promotion of Allergic Sensitization via the Skin Epithelium

In allergic sensitization, the first ‘defense’ against allergens is epithelial cells, either as skin cells, lung cells or intestinal cells. EVs generated from exogenous sources, such as bacteria, must thus cross this first barrier in order to interact with the immune system and influence the allergic sensitization process. Publications suggesting that exogenous EVs can also promote sensitization resulting in allergies, were identified (Table 7).

Five studies showed that EVs generated by bacteria could damage the skin barrier, leading to easier colonization of the skin, which in turn may influence allergic sensitization and AD [33,34,35,36,37]. Five studies were experimenting with *S. aureus* EVs and their influence on the induction of atopic dermatitis-like skin inflammation [33,34,35,36,37]; however, one paper showed that Lactobacillus plantarum-derived EVs could prevent damage to the skin from *S. aureus* EVs [36]. In one of these studies, the toxin α-Hemolysin produced by *S. aureus* was more potent in EV form and could induce IL-6 production associated with Th17 response, which caused skin barrier disruption and AD-like skin inflammation [33]. After investigating the potential effects of methicillin-resistant *Staphylococcus aureus* (MRSA) EVs in mice, it was found that the EVs act as immunostimulant that induces inflammatory responses, producing cytokines such as IL-4, IL-5 and IL-6 and triggering IgE-mediated hypersensitivity after MRSA infection [34].

The isolation protocols for the five studies were similar, where filtration and serial ultracentrifugation were used [33,34,35,36,37], which is not efficient enough to provide pure and intact EVs [38]. One study used filtration and an ExoQuickTC kit that utilized SEC [37]. The identification methods used in most studies were also not comprehensive, where 3 studies only used one method of identification for characterizing their EVs [35,36,37], while only one study utilized novel techniques on top of the standard methods [34]. Papers in external sources scored higher than the other sections in this systematic review. The highest scoring paper was 79% [34], due to the use of a novel identification method utilizing electrolytes in NP150 nanopore membranes on top of other conventional techniques such as Western blots, SDS-page gels and Bradford protein assays. The lowest scoring paper was also found in this section, with 38% [35], because of the poor isolation and characterization of their EVs, plus the use of only a murine in vivo model. The full assessment of papers can be found in Table 8.

Overall, in these five studies, bacterial EVs were shown to damage the epithelial cell barrier, which can lead to increased susceptibility to developing an allergic reaction.

#### 3.3.2. Promotion of Immune Tolerance

It has been shown that immune cells can recognize and interact with these exogenous EVs to alter their responses to stimuli [39]. Only one study showed that exogenous EVs promote immune tolerance (Table 9). Observing that EVs produced by Gram-negative bacteria in indoor dust can induce neutrophilic pulmonary inflammation and increased production of IFN-γ and IL-17 associated with both Th1 and Th17 cell response. Furthermore, dust EVs from Gram-negative bacteria induced tumor necrosis factor alpha (TNF-α) and IL-6 production in macrophages [40].

The one study showing that an exogenous source of EVs can tolerize the immune system used a mice model and a human in vitro model [40]. The isolation method utilized in this paper was ultracentrifugation [40]. The paper used NTA to size the EVs and used TEM [40] to confirm the morphology of the EVs. This paper scored well at a 58% bias score, as it utilized multiple model systems and fully described them [40]. A full breakdown of the scoring can be found in Table 10.

In summary, this paper shows that exogenous-sourced EVs were capable of shifting the immune response towards a Th1-like response.

## 4. Discussion

Allergies are a worldwide issue with an increasing number of cases in the population [41], yet the mechanisms behind the development of an allergy are still poorly understood. However, recently, there has been a focus on research on EVs and their role in allergies and their development. In this systematic review, we highlight the increasing evidence that EVs can promote allergic sensitization; conversely, there is also much evidence for their role in promoting tolerance to allergens.

Of the 22 studies relevant to EVs in allergic sensitization, 64% of studies [17,18,19,20,21,22,23,33,34,35,36,37,40] showed evidence of EVs enhancing the allergic sensitization process and shifting towards Th2-type responses, compared to 36%, which reported EVs could prevent allergic sensitization and shift the immune response towards a Th1 and Treg-like immune tolerance [24,25,26,28,29,30,31,32]. In terms of sensitization, this systematic review highlights that EVs promote the differentiation of naïve T cells into Th2 cells by interacting with immune cells to influence cytokine production, such as increasing IL-4, IL-13 and IL-5 concentration and decreasing IFN-γ levels [17,18,19,20,21,22,23]. In terms of tolerance, we highlight that EVs can induce immune tolerance by stimulating Th1 and Treg differentiation by increasing concentrations of TGF-β, IL-2 and IL-10 cytokines while suppressing the production of IL-4, IL-9, and IL-13 cytokines, leading to lower levels of Th2 differentiation [24,25,26,28,29,30,31,32]. In addition, this review highlights the origin of EV impacts, whether the EV promotes tolerance or sensitization, where EVs generated internally by the body influence both Th1 and Th2 responses, but exogenous EVs generated by bacteria mainly promote allergen exposure by increasing the permeability of the skin, thus resulting in increased chance of allergens interacting with immune cells [33,34,35,36,37,40]. Furthermore, bacteria-derived EVs can act as the ‘allergen’ and trigger allergic sensitization itself by stimulating IL-4, IL-5, and IL-6 production, and then, on re-exposure, trigger an IgE-mediated response [34]. 

One of the key mechanisms driving allergic sensitization suggested is EVs produced by DCs, B cells, Tfh, and epithelial cells that can affect the cytokine production in immune cells by promoting IL-4, IL-5, and IL-13 production while simultaneously decreasing the levels of IFN-γ and IL-17. For example, DC-derived EVs induced Th2 differentiation and suppressed Th1 responses, which can lead to allergic sensitization [17,18,19,20,21,23]. Currently, what induces naïve T cells to differentiate into Th2 cells during allergic sensitization is poorly understood [42]; however, the evidence presented here shows that EVs generated by immune cells can act as the signal to trigger the Th2 differentiation. Further research in this area could lead to developing new treatments to stop naïve—cell differentiation into Th2 cells, thus stopping allergic sensitization. Conversely, it is possible to reduce Th2 responses by inducing ‘desensitization’ by immunotherapy, where allergen-specific IgG, more specifically IgG4, antibodies are produced and compete with IgE and show inhibitory capacity for IgE-dependent events [43]. IgG4 can bind to the allergen specifically, thus blocking it from interacting with IgE, meaning decreased levels of IgE-allergen deposits on mast cells and, in turn, decreased histamine levels.

Furthermore, it has been shown that IgG4 competition can block IgE-allergen complexes binding to B cells, thereby inhibiting IgE-facilitated antigen presentation to T cells, which is a major driving force for generating allergen-specific Th2 responses [44]. Recent research has been exploring the potential of EVs used in therapies utilizing the ‘desensitization’ process [45]. Another mechanism by which EVs can promote allergic sensitization is by damaging the epithelial barrier by carrying lncRNA NEAT1, which interacts with HNECs to promote IL-13 and apoptosis, thus enhancing the likelihood of allergen exposure to immune cells [22].

In addition, a study by Gon et al. found that EVs released in the bronchoalveolar lavage fluid (BALF) were enriched with miRNAs once exposed to house-dust mites, and these EVs could upregulate Th2 cytokines [46], highlighting the importance of EVs as carriers of effector molecules. Furthermore, a study found key evidence where bronchial epithelial cells secreted by EVs carry a pivotal protein tissue factor (TF) in asthma and are involved in its pathogenesis [47]. This highlights the importance of studying the contents of EVs, as their cargo can be essential in their function.

As aforementioned, it is also possible for EVs to induce Th1 and Treg responses, thus inducing immune tolerance towards the allergen. One of the key mechanisms suggested is inducing Tregs in the immune system by increasing the production of TGF-β [24,25,26,28,29,31,32]. Furthermore, it is suggested that EVs carrying integrin αβ6 and OVA could interact with T cells and induce Treg differentiation, thus suppressing allergic sensitization [24].

Another mechanism of suppression of Th2 differentiation was by HMSC-derived EVs influencing the miR-146a-5p/SERPINB2 pathway in healthy CD4+ T cells [30], where it can be speculated that EVs can affect intrinsic pathways in the cells that they interact with either by activating the recipient cells receptors, thus initiating intracellular signaling pathways or cargo by fusion with the plasma membrane of target cells [48]. In addition, previously published literature has found how EVs can affect specific cells through intrinsic signaling pathways; for example, a study done by Salem et al. showed that EVs generated from bronchial fibroblasts could promote epithelial cell line remodeling through the TGF-β2 signaling pathway in severe asthma [49], suggesting that EVs are affecting the target cells’ intrinsic pathways and thus altering their function.

Another study found that EVs can also induce immune tolerance towards unrelated allergens, as well as specific allergens, by limiting the production of IL-5 and IL-13, thus inducing ‘bystander tolerance’ [32]. 

Recently, there has been more focus on the microbiome of humans and how it can affect the immune system [50]. It was found that EVs produced by bacteria, specifically *S. aureus*, could be the leading cause of damage to the skin and induction of AD-like symptoms, which are most likely caused by the EV cargo. For instance, the toxin α-Hemolysin was found to be more potent in EVs than in soluble form, causing further skin damage [33,35,36,37,40]. Damaged skin barrier can lead to increased susceptibility to allergic sensitization of allergens. This is due to the compromised barrier releasing defensins, which can predispose to allergic sensitization by subsequent interactions with T cells and can thus lead to atopic dermatitis [51]. In this systematic review, a major function of EVs was suggested by a study regarding MRSA-derived EVs, where EVs acted as immunostimulants, inducing IL-4, IL-5, and IL-6 cytokines in mice. Furthermore, following subsequent exposure to MRSA, an IgE-mediated hypersensitivity developed [34]. This suggests that EVs were able to sensitize the mice towards MRSA, directly suggesting that EVs can trigger allergic sensitization without additional signals from the mice.

It is worth noting that many of the studies examined had low-quality assessment scores, as many of the included studies had poor methods of EV isolation and characterization. The papers providing evidence for host-derived EVs and their influence on allergic sensitization scored a mean value of 57%, which was minimally higher by 5% than host-derived EVs inducing tolerance, showing the reliability of the key mechanisms was similar in these sections. However, the highest scoring section was exogenous EVs in promoting allergic sensitization with a 62% mean score, while exogenous EVs and tolerance had 58% scoring quality, indicating the robustness of these studies was of similar quality. As the sections do not have high variance between their scoring, there is no indication of higher reliability in the findings between the sections. The quality scoring highlights the need for further understanding of EVs and how to isolate them more efficiently to increase purity and maintain their original form without rupture. Only two studies [19,21] received the highest points for EV characterization. In addition, five different studies [20,28,30,33,37] had the highest score of EV isolation. Thus, future research must utilize new isolation techniques to ensure the highest purity of EVs and the exclusion of cytokines, as it has been shown that cytokine presence can affect the function of the EVs.

Furthermore, fully characterizing EVs would lead to a better understanding of the differences between the sub-populations of EVs and which receptors, and what cargo, drives their role in allergy. In addition, almost all studies lacked human ex vivo experiments, except one study [37], which used a foreskin puncture to show a more accurate representation of the immune and epithelial cell interactions. Interestingly, almost all studies had high scores regarding the robustness of the model, as they clearly described what cell type was used, and the duration, exposure method, and dosing of the allergen were fully explained. All the studies received the same score for material used for sensitization as the allergen was clearly described and measured; however, none of the studies performed endotoxin measurement to ensure the allergen was endotoxin-free.

This systematic review contained studies both using human and murine models, with 57% of studies utilizing murine models for the study of allergic sensitization. However, the use of mice in allergy studies is not preferred, as an allergy in mice is induced artificially and does not fully represent allergic sensitization in humans. In addition, mice have been shown to produce cytokines differently than humans; for example, IL-10 in mice is only produced by Th2 response, while in humans, it can be both Th1 and Th2 responses [52].

Future research in the field of EVs could open avenues for understanding the development of allergic sensitization, possibly leading to the development of new treatments and/or prevention of allergic sensitization. It is clear EVs play a role in allergies, either in their development or by induction of tolerance, and this is dependent on the cell of origin, health status, and environmental conditions in which the cells grow. In turn, this can influence the outcome of the EVs generated, resulting in different cargo encapsulations and differing cellular outcomes [53]. In the future, more research is required on EVs released by cells, investigating their cargo and proteins on the surface, which interact with immune cells to shift the response away from allergic sensitization and towards tolerance.

## 5. Conclusions

In summary, this systematic review provides evidence for the role of extracellular vesicles in allergies, where they have been shown to influence Th1, Th2 or Treg responses to allergens. This evidence suggests that EVs could be the driving force that decides the immune response to an allergen. However, with the limited number of papers that have been published, further research is crucial in fully understanding the role of EVs in allergies.

## Figures and Tables

**Figure 1 ijms-25-04492-f001:**
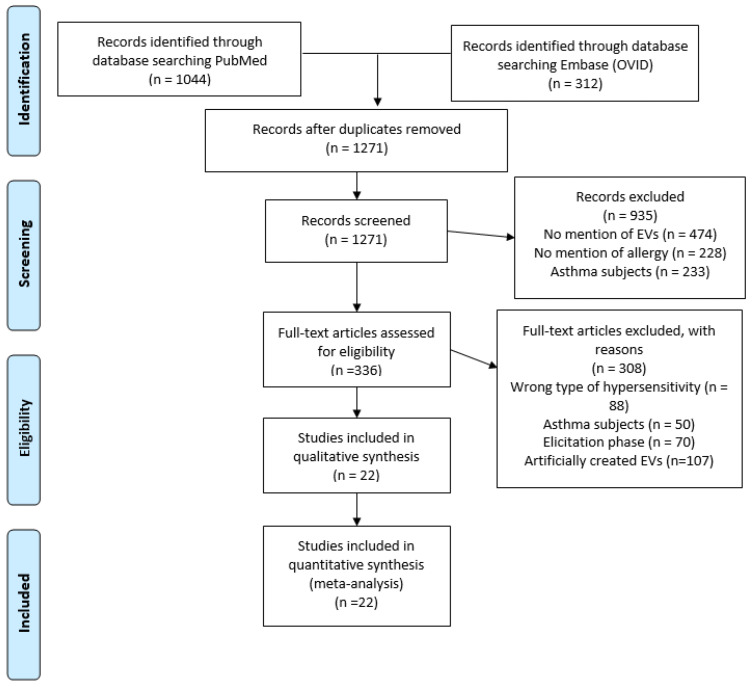
PRISMA diagram of article search and selection. A PRISMA diagram detailing the process of searching and selecting EV studies in allergic sensitization, using PubMed and Embase (OVID) from 1974–2023.

**Figure 2 ijms-25-04492-f002:**
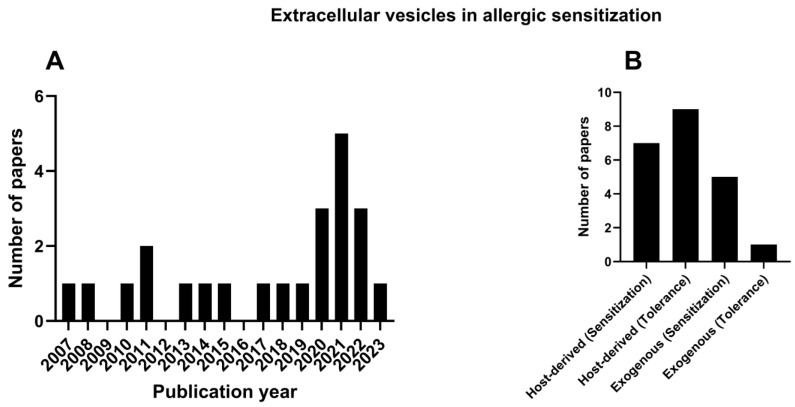
Publication on EVs in allergic sensitization through the years (**A**). Publications concerning the host-derived EVs—sensitization and tolerance, and exogenous source EVs (**B**).

**Table 1 ijms-25-04492-t001:** Selection criteria for papers included in the systematic review.

**Inclusion**	**Exclusion**
IgE-mediated food allergy	Asthma
IgE-mediated inhalant allergy	Non-IgE-mediated allergies
Allergic sensitization	Non-English language publications
English language	Elicitation phase
Full-text available	Reviews
Experimental data	Conference abstracts
Healthy subjects	Contact hypersensitivity
Allergic subjects	Delayed hypersensitivity
Human models	Artificially created EVs
Animal models	EVs used as biomarkers
	Therapeutic use of EVs

**Table 2 ijms-25-04492-t002:** Scoring system to assess the robustness of included papers.

Category	Reasoning for Scores	
Model	If multiple models are used, then a combined score is given: - Human (ex vivo) (4); - Cell Culture (in vitro—human) (3); - Murine Model (in vivo) (2); - Cell Culture In vitro—murine) (1).	
Robustness of model	If multiple models are used, then a combined score is given: Human model (ex vivo): - Allergic status not specified or not clearly defined (0); - Allergic patients were sought from a Clinical Setting (1); - Allergic patients were sought from the Clinical Setting AND had positive testing for an allergen (Specific IgE or Positive Allergen Challenge) (2); - Allergic patients were sought from a Clinical Setting with a healthy control group (3); - Allergic patients were sought from a Clinical Setting AND had a positive test for an allergen (Specific IgE or Positive Allergen Challenge) with a healthy control group (4). Murine Model (in vivo): - Sensitization of mice not specified or defined (0); - Sensitization of mice partially defined (discloses some but NOT all the following: exposure method, dosing, and duration) (1); - Sensitization of mice fully defined outlining (exposure method, dosing, and duration) (2); - Sensitization of mice partially defined (Discloses some but NOT all the following: exposure method, dosing, and duration) with a negative control group (3); - Sensitization of mice fully defined outlining (exposure method, dosing, and duration) with a negative control group (4), Cell Culture (in vitro): If both are used, the greater model score will be used. - Immortalized Cell Line using partially defined exposures (Discloses some but NOT all the following: exposure method, dosing, and duration) (1); - Immortalized Cell Line using fully defined exposures (exposure method, dosing, and duration) (2); - Primary Cell Line using partially defined exposures (discloses some but NOT all the following: exposure method, dosing, and duration) (3); - Primary Cell Line using fully defined exposures (exposure method, dosing, and duration) (4).	
Sample size	If multiple models are used, then a combined score is given. Human model (ex vivo): - Number of participants not defined (0); - 5 or less participants per group (1); - 6 to 10 participants per group (2); - 11 or more participants per group (3). Murine Model (in vivo): - Number of mice per group not defined (0); - 5 or less mice per group (1); - 6 to 10 mice per group (2); - 11 or more mice per group (3). Cell Culture (in vitro): - n not specified (0); - n = 1 (1); - n = 3 (2); - n > 3 (3);	
EV isolation	- No Isolation—Techniques using only precipitation. Studies looking directly in a liquid with no isolation applied (usually done on very low volume samples) (0); - Poor—Ultracentrifugation (UC) at one speed or serial UC without sucrose cushion (1); - Fair—Size exclusion chromatography (without a precipitation or concentration step), immuno-capture beads for investigation of non-specific populations (2); - Good—Size exclusion chromatography (SEC) with a precipitation or concentration step or the use of a specific isolation method such as exosome EV isolation, purification kits or immuno-capture beads utilizing a unique marker (3).	
EV characterization	- No Characterization—No attempt made to profile or characterize EVs or Exosomes (0); - Poor—Use of just one characterization technique (quantification, sizing, biomarkers, or cargo analysis) (1); - Fair—The use of multiple complementary characterization techniques and at least 1 biomarker (2); - Good—Everything mentioned prior, as well as appropriate controls (e.g., robust profiling of culture conditions such as media, the inclusion of positive and negative controls such as those recommended by MISEV2018) (3); - Very Good—Everything mentioned prior as well as the addition of extra biomarkers (3+) or the utilization of any other novel techniques (4).	
Overall quality score	The combined score of the categories above is divided by the highest possible score depending on the model used (24/31/37).	

**Table 3 ijms-25-04492-t003:** Studies involving allergic sensitization and host-derived EVs.

Author (Ref.)	Title	Year	Objective	Isolation Technique	Identification of EVs	Findings
Admyre et al., 2007 [17]	B-Cell-Derived Exosomes Can Present Allergen Peptides and Activate Allergen-Specific T Cells to Proliferate and Produce TH2-Like Cytokines	2007	Role of antigen-presenting cell-derived exosomes in allergen presentation and T-cell stimulation.	Ultracentrifugation—300× *g*, 3000× *g* 20 min, 10,000× *g* 30 min at 4 °C, 110,000× *g* 1 h at 4 °C.	Adsorbed onto 4.5 μm Dynabeads precoated with anti-MHC class 2. (enzyme-linked immunosorbent assay (ELISA).	B-cell-derived exosomes can present allergen-derived peptides and thereby induce T-cell proliferation and TH2-like cytokine production.
Fang et al., 2021 [18]	Plasma EVs Display Antigen-Presenting Characteristics in Patients With Allergic Rhinitis and Promote Differentiation of Th2 Cells	2021	Role of plasma EVs in allergic rhinitis.	Differential ultracentrifugation—600× *g* 10 min, 2000× *g* 20 min, 12,000× *g* 30 min, 110,000× *g* 2 h and 70 min.	ELISA, nanoparticle tracking analysis (NTA), transmission electron microscopy (TEM), and Western blot for EV markers. CSFE stained EVs in flow cytometry.	Plasma EVs derived from patients with AR exhibited antigen-presenting characteristics and promoted differentiation of Th2 cells.
Huang et al., 2018 [19]	Exosomes from Thymic Stromal Lymphopoietin-Activated Dendritic Cells Promote Th2 Differentiation through the OX40 Ligand	2019	Role of exosomes from DCs activated by (TSLP) in T-helper cell differentiation through the OX40 ligand.	Ultracentrifugation of DC S/Ns, re-suspended in PBS and stored at –80 °C.	TEM to observe, Bradford dye assay for protein concentration, Western blot for CD63.	TSLP-activated DCs produce OX40L and sOX40L and EVs could transport it. TSLP-DEXs were promoting Th2 response in CD4+ T cells. Blockade of OX40L in DC-derived exosomes could inhibit exosome-mediated CD4+ T proliferation and Th2 differentiation.
Teng et al., 2022 [20]	Tfh Exosomes Derived from Allergic Rhinitis Promote DC Maturation Through miR-142-5p/CDK5/STAT3 Pathway	2022	To explore the mechanism of Tfhs on DC. Maturation in AR.	Ultracentrifugation 100,000× *g* 90 min.	TEM	AR-Tfh-exos promoted TH2 respones in naïve mice, promoting both nasal inflammation and affecting naïve T cells.
Vallhoy et al., 2015 [21]	Dendritic Cell-Derived Exosomes Carry the Major Cat Allergen Fel d 1 and Induce an Allergic Immune Response	2015	Whether DC-derived exosomes can present the major cat allergen Fel d 1 and whether they contribute to the pathogenesis of allergic disease.	Serial ultracentrifugation 90,000× *g* 90 min.	iEM with gold particles. NTA. Sulfate-aldehyde latex microspheres labeled with CD63 and CD81 run on FACSCalibur. ELISA and TEM.	Exosomes can present aeroallergens and thereby induce T-cell T(H)2-like cytokine (IL-4) production in allergic donors.
Wang et al., 2021 [22]	Exosomal lncRNA Nuclear Paraspeckle Assembly Transcript 1 (NEAT1) Contributes to the Progression of Allergic Rhinitis via Modulating MicroRNA-511/Nuclear Receptor Subfamily 4 Group A Member 2 (NR4A2) Axis	2021	Functions and molecular mechanisms of NEAT1 in AR.	ExoQuick precipitation kit, centrifuged 3000× *g* 15 min, 30 min at 4 °C, then 1.5 k 30 min at 4 °C.	Western blot for CD9 and CD63, TEM	Exosomal NEAT1 contributed to the pathogenesis of AR through the miR-511/NR4A2 axis. Exposing HNECs with AR-induced HNECs-derived exosomes resulted in inflammatory responses.
Zhu et al., 2020 [23]	Exosomal Long Non-Coding RNA GAS5 Suppresses Th1 Differentiation and Promotes Th2 Differentiation via Downregulating EZH2 and T-bet in Allergic Rhinitis	2020	Role of LncGAS5 in EVs in T-cell responses.	Ultracentrifugation—12,000× *g* 45 min at 4 °C, 110,000× *g* 2 h at 4 °C, filtered 0.22 qm, then 110,000× *g* 70 min at 4 °C.	TEM, Western Blot for CD63 and CD81.	LncGAS5 in AR epithelium-derived exosomes is the key mediator in Th1/Th2 differentiation. OVA-EXO suppresses Th1 differentiation and promotes Th2 differentiation.

**Table 4 ijms-25-04492-t004:** Scoring assessment of papers involving host-derived EVs and allergic sensitization.

Reference	Model (n/6)	Robustness of Model		Sample Size		Sensitization (n/3)	EV Isolation (n/3)	EV Characterization (n/4)	Total Score (n/27) or (n/34)	Bias Score
		Murine Model (n/4)	Cell Culture (n/4)	Murine Model (n/3)	Cell Culture (n/3)					
Admyre et al., 2007 [17]	Human (in vitro) (3)		Fully defined exposures (4)		More than 3 repeats (3)	Allergen defined, its conc. measured (2)	Ultracentrifugation (1)	One method used (1)	14/27	52%
Fang et al., 2021 [18]	Human (in vitro) (3)		Fully defined exposures (4)		More than 3 repeats (3)	Allergen defined, its conc. measured (2)	Ultracentrifugation (1)	Multiple complimentary techniques, plus a novel technique utilized (4)	17/27	63%
Huang et al., 2018 [19]	Human (in vitro) (3)		Fully defined exposures (4)		3 repeats (2)	Allergen defined, its conc. measured (2)	Ultracentrifugation (1)	Multiple complimentary techniques (2)	14/27	52%
Teng et al., 2022 [20]	Human (in vitro) and murine (in vivo) (5)	Fully defined mice sensitization (4)	Fully defined exposures (4)	5 or less mice (1)	3 repeats (2)	Allergen defined, its conc. measured (2)	Ultracentrifugation (1)	One method used (1)	21/34	62%
Vallhoy et al., 2015 [21]	Human (in vitro) (3)		Fully defined exposures (4)		More than 3 repeats (3)	Allergen defined, its conc. measured (2)	Ultracentrifugation (1)	Multiple complimentary techniques, plus a novel technique utilized (4)	17/27	63%
Wang et al., 2021 [22]	Human (in vitro) (3)		Fully defined exposures (4)		3 repeats (2)	Allergen defined, its conc. measured (2)	Exoquick isolation kit (size chromatography) (3)	One method used (1)	15/27	56%
Zhu et al., 2020 [23]	Human (in vitro) (3)		Fully defined exposures (4)		More than 3 repeats (3)	Allergen defined, its conc. measured (2)	Ultracentrifugation (1)	One method used (1)	14/27	52%

**Table 5 ijms-25-04492-t005:** Studies in host-derived EVs and allergic tolerance.

Author (Ref.)	Title	Year	Objective	Isolation Technique	Identification of EVs	Findings
Chen et al., 2011 [24]	Intestinal Epithelial-Cell-Derived Integrin αβ6 Plays an Important role in the Induction Of Regulatory T Cells and Inhibits an Antigen-Specific Th2 Response	2011	Role of integrin αβ6 and EVs in T cell differentiation.	Serial centrifugation—300× *g* 10 min, 1200× *g* 20 min, 10,000× *g* 30 min, 100,000× *g* 1 h	Bradford assay, TEM.	In vivo administration of αvβ6/OVA-laden exosomes induced the generation of Tregs and suppressed skewed Th2 responses toward food antigens in the intestine.
Yu et al., 2020 [25]	Specific Antigen-Guiding Exosomes Inhibit Food Allergies by Inducing Regulatory T Cells	2020	To suppress experimental food allergy (FA) by inducing Tregs through the employment of modified exosomes (mExosomes).	Serial ultracentrifugation—300× *g* 10 min, 1200× *g* 20 min, 10,000× *g* 30 min, 100,000× *g* for 1 h.	Bradford assay.	Administration of mExosomes induced Tregs in the intestinal tissues and efficiently suppressed FA in mice.
Zeng et al., 2020 [26]	Exosomes Carry IL-10 and Antigen/MHC II Complexes to Induce Antigen-Specific Oral Tolerance	2020	To investigate the role of vasoactive intestinal peptide (VIP) in the immune tolerance development in the intestine.	Ultracentrifugation—300× *g* 10 min, 1200× *g* 20 min, 10,000× *g* 30 min, 100,000× *g* 1 h, CD9 coated magnetic beads.	Bradford assay.	IL10CARs (produced from IEC exposed t VIP and OVA) induced Treg in healthy CD4 T cells, thus suppressing FA.
Asadirad, et al. 2023 [27]	Sublingual Prophylactic Administration of OVA-Loaded MSC-Derived Exosomes to Prevent Allergic Sensitization	2023	Adipose tissue-isolated MSC-derived exosomes as a prophylactic regimen through a sublingual route in the ovalbumin (OVA)-induced allergic murine model.	Exospin isolation kit (SEC)	NTA, SEM, immunophenotyping of CD9 and C63.	Significant reduction in the IgE levels and IL-4 production, along with elevated TGF-β levels, were observed. Also, limited cellular infiltrations, perivascular and peribronchiolar inflammation in the lung tissues, and normal total numbers of cells and eosinophils in the nasal lavage fluid (NALF) were reported.
Peng et al., 2022 [28]	Mesenchymal Stromal Cell-Derived Small Extracellular Vesicles Modulate DC Function to Suppress Th2 Responses via IL-10 in Patients with Allergic Rhinitis	2022	MSC-derived small extracellular vesicles (MSC-sEV) effects on DCs in allergic diseases.	2000× *g* 20 min at 4 °C, anion exchange chromatography using Econo-Pac column, then concentrated using a Pierce Protein Concentrator.	Bradford assay, NTA, TEM, Western blotting.	The paper identified that sEV-mDCs suppressed the Th2 immune response by reducing the production of IL-4, IL-9, and IL-13 via IL-10. Furthermore, sEV-mDCs increased the level of Treg cells.
Prado et al., 2008 [29]	Exosomes from Bronchoalveolar Fluid of Tolerized Mice Prevent Allergic Reaction	2008	The effect of allergen-specific exosomes from tolerized mice on the development of allergen-induced allergic response was determined using a mouse model.	Ultracentrifugation.	TEM, microbicinchonic acid assay, SDS-PAGE and Western blotting, coated beads on FACS.	Bronchoalveolar lavage fluid (BALF)-derived Exotol inhibits IgE and induces IgG2a cytokine production. These observations demonstrate that exosomes can induce tolerance and protection against allergic sensitization in mice.
Zhou et al., 2021 [30]	HMSC-Derived Exosome Inhibited Th2 Cell Differentiation via Regulating miR-146a-5p/SERPINB2 Pathway	2021	The study aimed to investigate the role of HMSC-exos in the pathogenesis of AR.	Exoquick, serial centrifugation—500× *g* 10 min, twice 2000× *g* 15 min twice 10,000× *g* 30 min, 70,000× *g* 1 h at 4 °C	Western blotting for CD63, CD81, TEM.	HMSC-exos could inhibit the differentiation of Th2 cells via the regulation of the miR-146a-5p/SERPINB2 pathway. miR-146a-5p and Serpin Family B Member 2 (SERPINB2) could be applied as potential targets for AR treatment.
Wahlund et al., 2017 [31]	Exosomes from Antigen-Pulsed Dendritic Cells Induce Stronger Antigen-Specific Immune Responses than Microvesicles In Vivo.	2017	To investigate both MVs and exosomes from Ovalbumin (OVA)-pulsed dendritic cells for their immunostimulatory potential side-by-side in vivo.	Differential centrifugations—300× *g* 10 min, 3000× *g* 30 min, 10,000× *g* for 40 min to get MVs, S/N 100,000× *g* for 90 min.	Transmission Electron Microscopy	DC-derived MVs and exosomes differ in their capacity to incorporate antigens and induce immune responses. Exosomes were more efficient in inducing CD8 T cells and IgG production than microvesicles.
Prado et al., 2010 [32]	Bystander Suppression to Unrelated Allergen Sensitization Through Intranasal Administration of Tolerogenic Exosomes in Mouse.	2010	To investigate whether nanovesicles specific to Ole e 1 can also prevent the sensitization to other unrelated allergen, such as Bet v 1 from birch pollen.	Filtration on 0.22 qm pores and ultracentrifugation at 100,000× *g* 1 h.	Protein conc measured by micro-bicinchoninic acid assay.	ExoTol specific to Ole e 1, in addition to inhibiting specific immune response to this allergen, blocked the allergic response to a second unrelated allergen such as Bet v 1. The in vivo “bystander suppression”.

**Table 6 ijms-25-04492-t006:** Scoring assessment of papers in host-derived EVs and allergic tolerance.

Author (Ref.)	Model (n/10)	Robustness of Model		Sample Size		Sensitization (n/3)	EV Isolation (n/3)	EV Characterization (n/4)	Total Score (n/27) or (n/34)	Bias Score
		Murine Model (n/4)	Cell Culture (n/4)	Murine Model (n/3)	Cell Culture (n/3)					
Chen et al., 2011 [24]	Murine (in vitro and vivo) (3)	Mice defined, no negative control (2)	Fully defined exposures (4)	5 or less mice (1)	3 repeats (2)	Allergen defined, its conc. measured (2)	Ultracentrifugation performed (1)	One method used (1)	16/34	47%
Yu et al., 2020 [25]	Murine (in vitro and vivo) (3)	Fully defined mice sensitization (4)	Fully defined exposures (4)	6 to 10 mice per group (2)	3 repeats (2)	Allergen defined, its conc. measured (2)	Ultracentrifugation performed (1)	One method used (1)	19/34	56%
Zeng et al., 2020 [26]	Murine (in vitro and vivo) (3)	Fully defined mice sensitization (4)	Fully defined exposures (4)	6 to 10 mice per group (2)	More than 3 repeats (3)	Allergen defined, its conc. measured (2)	Ultracentrifugation performed (1)	One method used (1)	20/34	59%
Asadirad et al., 2023 [27]	Murine (in vivo) (2)	Fully defined mice sensitization (4)			5 or less mice (1)	Allergen defined, its conc. measured (2)	Exospin kit (SEC) (3)	Multiple complimentary techniques (2)	14/34	41%
Peng et al., 2022 [28]	Human (in vitro) (3)		Fully defined exposures (4)		More than 3 repeats (3)	Allergen defined, its conc. measured (2)	Anion exchange chromatography (3)	Multiple complementary techniques (2)	17/27	63%
Prado et a., 2008 [29]	Murine (in vivo) (2)	Fully defined mice sensitization (4)		5 or less mice (1)		Allergen defined, its conc. measured (2)	Ultracentrifugation performed (1)	Multiple complementary techniques (2)	12/27	44%
Zhou et al., 2021 [30]	Human (in vitro) (3)		Fully defined exposures (4)		3 repeats (2)	Allergen defined, its conc. measured (2)	Exoquick isolation kit (size chromatography) (3)	One method used (1)	15/27	56%
Wahlund et al., 2017 [31]	Murine (in vitro and vivo) (3)	Fully defined mice sensitization (4)	Fully defined exposures (4)	5 or less mice (1)	3 repeats (2)	Allergen defined, its conc. measured (2)	Ultracentrifugation performed (1)	One method used (1)	18/34	53%
Prado et al., 2010 [32]	Murine (in vitro and vivo) (3)	Fully defined mice sensitization (4)	Fully defined exposures (4)	5 or less mice (1)	Less than 3 repeats (1)	Allergen defined, its conc. measured (2)	Ultracentrifugation performed (1)	One method used (1)	17/34	50%

**Table 7 ijms-25-04492-t007:** Studies involving external EVs and allergic sensitization.

Author (Ref.)	Title	Year	Objective	Isolation Technique	Identification of EVs	Findings	
Hong et al., 2014 [33]	An Important Role of α-Hemolysin in Extracellular Vesicles on the Development Of Atopic Dermatitis Induced by *Staphylococcus aureus*	2014	The role of α-hemolysin in EVs on the development of atopic dermatitis induced by *Staphylococcus aureus.*	Filtered, ultracentrifugation 150,000× *g* 3 h.	Bicinchoninic acid assay (BCA) assays, confocal microscopy, Western blot.	α-Hemolysin secreted from *S. aureus*, particularly the EV-associated form, induces both skin barrier disruption and AD-like skin inflammation, suggesting that EV-associated α-hemolysin is a novel diagnostic and therapeutic target for the control of atopic dermatitis (AD).	
Asano et al., 2021 [34]	Extracellular Vesicles from Methicillin-Resistant *Staphylococcus aureus* Stimulate Proinflammatory Cytokine Production and Trigger IgE-Mediated Hypersensitivity	2021	Role of EVs from a clinically isolated methicillin-resistant *S. aureus* (SaEVs) on the immune system.	Ultracentrifuge—5,000× *g* 20 in, 100,000× *g* 90 min at 4 °C, purified using OptiPrep density gradient, 100,000× *g* 16 h at 4 °C.	TEM, qNano, Measurmenet Electrolytes used in NP150 nanopore membranes. SD-Page, Bradford protein assays, Western blot.	MRSA-derived EVs act as an immunostimulant that induces inflammatory response and IgE-mediated hypersensitivity after MRSA infection.	
Hong et al., 2011 [35]	Extracellular vesicles derived from *Staphylococcus aureus* induce atopic dermatitis-like skin inflammation	2011	The study evaluates whether *S. aureus*-derived EVs are causally related to the pathogenesis of AD.	Ultracentrifugation 150,000× *g*. Specifically, 0.45 µm vacuum filtration, concentrated by QuixStand Benchtop System, then 0.22 µm vacuum filter before ultracentrifugation.	Bradford assays, ELISA	*S. aureus* EVs induce AD-like inflammation in the skin, and *S. aureus*-derived EVs are a novel diagnostic and therapeutic target for the control of AD.	
Kim et al., 2018 [36]	Lactobacillus plantarum-derived Extracellular Vesicles Protect Atopic Dermatitis Induced by Staphylococcus aureus-derived Extracellular Vesicles	2018	The study aims to compare the bacterial EV composition between AD patients and healthy subjects and to experimentally find out the beneficial effect of some bacterial EV composition.	Multiple ultracentrifugation and purification processes	TEM, SDS-Page, Zetasizer Nano ZS	The study suggests the protective role of lactic acid bacteria in AD based on metagenomic analysis. Experimental findings further suggested that L. plantarum-derived EV could help prevent skin inflammation	
Staudenmaier et al., 2022 [37]	Bacterial membrane vesicles shape *Staphylococcus aureus* skin colonization and induction of innate immune responses	2021	Membrane vesicles (MVs) are released by pathogenic bacteria and might play an essential role in the long-distance delivery of bacterial effectors, such as virulence factors.	Size exclusion chromatography. Specifically, filtered, 3000× *g* for 30 min, ExoQuickTC, 1500× *g* for 30 min.	Western blotting	The data underlined the complex interplay in host- and bacterial-derived factors in *S*. *aureus* skin colonization and the important role of bacterial-derived MVs and their membrane lipid and protein A content in skin inflammatory disorders.	

**Table 8 ijms-25-04492-t008:** Scoring assessment of studies in exogenous EVs and allergic sensitization.

Reference	Model (n/10)	Robustness of Model			Sample Size			EV Isolation (n/3)	EV Characterization (n/4)	Total Score (n/24) (n/31) (n/38)	Bias Score
		Murine Model (n/4)	Cell Culture (n/4)	Human Model (n/4)	Murine Model (n/3)	Cell Culture (n/3)	Human Model (n/3)				
Hong et al., 2014 [33]	Human (in vitro) and Murine (in vivo) (5)	Fully defined mice sensitization (4)	Fully defined exposures (4)		5 or less mice (1)	3 repeats (2)		Ultracentrifugation (1)	Multiple complimentary techniques (2)	19/31	61%
Asano et al., 2021 [34]	Murine (in vitro and vivo) (3)	Fully defined mice sensitization (4)	Fully defined exposures (4)		5 or less mice (1)	3 repeats (2)		Ultracentrifugation (1)	Multiple complimentary techniques, plus a novel technique utilized (4)	19/24	79%
Hong et al., 2011 [35]	Murine (in vivo) (2)	Fully defined mice sensitization (4)			5 or less mice (1)			Ultracentrifugation (1)	One method used (1)	9/24	38%
Kim et al., 2018 [36]	Human (in vitro) and Murine (in vivo) (5)	Fully defined mice sensitization (4)	Fully defined exposures (4)		5 or less mice (1)	3 repeats (2)		Ultracentrifugation (1)	Multiple complimentary techniques (2)	19/31	61%
Staudenmaier et al., 2022 [37]	Human (Ex vivo and in vitro) (7)		Fully defined exposures (4)	Fully defined exposures (4)		3 repeats (2)	5 or less participants (1)	Exoquick isolation kit (size chromatography) (3)	One method used (1)	22/31	71%

**Table 9 ijms-25-04492-t009:** Studies involving external EVs and allergic tolerance.

Author (Ref.)	Title	Year	Objective	Isolation Technique	Identification Method	Findings
Kim et al., 2013 [40]	Extracellular vesicles, especially derived from Gram-negative bacteria, in indoor dust induce neutrophilic pulmonary inflammation associated with both Th1 and Th17 cell responses	2013	To evaluate whether EVs in indoor air are related to the pathogenesis of pulmonary inflammation and/or asthma.	10,000× *g* 15 min, 0.45 qm vacuum filter and concentrated using ultrafiltration QuixStand Benchtop System, filtered 0.22 µm vacuum filter, 150,000× *g* 3 h at 4 °C	TEM, NTA	In summary, the present data indicate that inhalation of indoor dust EV induces both Th1 and Th17 cell responses and neutrophilic inflammation in the lung.

**Table 10 ijms-25-04492-t010:** Assessment of studies involving exogenous EVs and allergic tolerance.

Reference	Model (n/10)	Robustness of Model		Sample Size		Sensitization (n/3)	EV Isolation (n/3)	EV Characterization (n/4)	Total Score (n/34) or (n/31)	Bias Score
		Murine Model (n/4)	Cell Culture (n/4)	Murine Model (n/3)	Cell Culture (n/3)					
Kim et al., 2013 [40]	Human (in vitro) and murine (in vitro and vivo) (6)	Fully defined mice sensitization (4)	Fully defined exposures (4)	5 or less mice (1)	3 repeats (2)	N/A	Ultracentrifugation (1)	One method used (1)	18/31	58%

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
