# Peer review of "The Role of Extracellular Vesicles in Allergic Sensitization: A Systematic Review"

_ijms, 2024, doi:10.3390/ijms25084492_

Round 1

Reviewer 1 Report

Comments and Suggestions for Authors

This nicely written systematic review by Tucis provides insight into the role of extracellular vesicles (EVs) in the development of allergy, and EV ability in shaping subsequent T-helper (Th) Th1, Th2 or T-regulatory responses. The References are up-to-date with summary findings listed as Table 3–9.

General comments:

1.     Please provide PRISMA study number in Figure 1.

2.     Dendritic cells (DCs), ie, Page 6 line 144. Any information on EVs and DC biology based on subset classification, ie, plasmacytoid DC (pDC), conventional (cDC 1&2), similar to the discussion based on T-cell subsets in this review.

3.     Discussion – it would be informative briefly mention EVs in vaccination, ie, by promoting Th2 in protective antibody (IgG) production for a more balanced view point rather than implying IgE through Type 1 hypersensitivity reaction by EV only.

Author Response

Thank you for the positive comments on our systematic review.  We have addressed the comments as follows:

General comments:

  1. Please provide PRISMA study number in Figure 1.

We followed the guidelines provided by MDPI and have included the PRISMA diagram, but have not undergone registration in PROSPERO, also as per MDPI suggested guidelines.

  1. Dendritic cells (DCs), ie, Page 6 line 144. Any information on EVs and DC biology based on subset classification, ie, plasmacytoid DC (pDC), conventional (cDC 1&2), similar to the discussion based on T-cell subsets in this review.

Thank you for your suggestion of including information about different subsets of DCs. This has now been added to the review.

  1. Discussion – it would be informative briefly mention EVs in vaccination, ie, by promoting Th2 in protective antibody (IgG) production for a more balanced view point rather than implying IgE through Type 1 hypersensitivity reaction by EV only.

Thank you for your suggestion of including relevant information on the production of IgG protective antibodies, this have now been added to the systematic review.

Reviewer 2 Report

Comments and Suggestions for Authors

This is an interesting review by D. Tucis et al. showing different original articles presenting evidence of EVs as important actors in the promotion of allergic sensitization or tolerance, depending on the cellular origin of internal EVs; whereas external (i.e. bacterial) EVs mainly induced allergic sensitization.

Comments:

.- The authors define a quality assessment scoring system to assess the robustness of the studies included in the review; but a clear utility of this scoring is not sufficiently described, neither in the results nor in the discussion section. For example, the text on lines 376-393 is only a summary of the results presented in the results section. Please explain in detail the usefulness of this score in the discussion section and use the information obtained with the score to explain the apparent discrepancies observed between the articles.

.- Information in the results section is sometimes redundant between tables and text. Please shorten this section.

.- Please, include the full explanation for each abbreviation used the first time it is given, and also use it below: e.g. methicillin resistant Staphylococcus aureus, MRSA; allergic rhinitis, AR (line 150); transmission electron microscopy, TEM (line 164); size-exclusion chromatography, SEC (line 260); etc.

.- Citation 15 (line 66) should be changed to the original articles citing these statements or instead be cited as “revised in 15”.

.- The sentence in lines 358-360 lacks citation(s).

.- Please review this sentence (line 155)  “…, where IFN-γ and Th1 concentrations were decreased…”. Would you mean “…, where IFN-γ concentrations and Th1 response were decreased…”?

.- Please avoid overuse of the expression “… in this systematic review”.

.- Please check and correct grammar and spelling errors: e.g. … EVs that were produced by from the host (Host-derived) or from external sources … (lines 122-123); on line 345, "Tr1" should be changed to "Th1".

Author Response

Thank you for the positive comments on our systematic review.  We have addressed the comments as follows:

Comments:

.- The authors define a quality assessment scoring system to assess the robustness of the studies included in the review; but a clear utility of this scoring is not sufficiently described, neither in the results nor in the discussion section. For example, the text on lines 376-393 is only a summary of the results presented in the results section. Please explain in detail the usefulness of this score in the discussion section and use the information obtained with the score to explain the apparent discrepancies observed between the articles.

Thanks for the insight of adding scoring differences between the sections of the review, this has now been added and enhances the key take-home messages of the paper.

.- Information in the results section is sometimes redundant between tables and text. Please shorten this section.

Thanks for highlighting the repetition in the results section. We have now reduced the text in tables in order to reduce repetition. We maintained the key findings in the body of the text to ensure that the main findings are not lost.

.- Please, include the full explanation for each abbreviation used the first time it is given, and also use it below: e.g. methicillin resistant Staphylococcus aureus, MRSA; allergic rhinitis, AR (line 150); transmission electron microscopy, TEM (line 164); size-exclusion chromatography, SEC (line 260); etc.

Thanks for pointing out the missing abbreviations. This has been addressed now both in the main body of the text and in the tables.

.- Citation 15 (line 66) should be changed to the original articles citing these statements or instead be cited as “revised in 15”.

Thank you for your suggestion of correctly referencing the systematic review. This has now been changed in the article

.- The sentence in lines 358-360 lacks citation(s)

Thank you for finding the missing citation, this has now been corrected.

Round 2

Reviewer 2 Report

Comments and Suggestions for Authors

Thank you very much for considering my comments. I think this review can be very useful for readers. Congratulations on your excellent work.